# Anti-*Helicobacter pylori* Activity of a *Lactobacillus* sp. PW-7 Exopolysaccharide

**DOI:** 10.3390/foods10102453

**Published:** 2021-10-14

**Authors:** Jingfei Hu, Xueqing Tian, Tong Wei, Hangjie Wu, Jing Lu, Mingsheng Lyu, Shujun Wang

**Affiliations:** 1Jiangsu Key Laboratory of Marine Bioresources and Environment, Jiangsu Key Laboratory of Marine Biotechnology, Jiangsu Ocean University, Lianyungang 222005, China; jfhu123@jou.edu.cn (J.H.); xqt@jou.edu.cn (X.T.); twei@jou.edu.cn (T.W.); wuhangjie@jou.edu.cn (H.W.); jinglu@jou.edu.cn (J.L.); mslyu@jou.edu.cn (M.L.); 2Co-Innovation Center of Jiangsu Marine Bio-Industry Technology, Jiangsu Ocean University, Lianyungang 222005, China

**Keywords:** *Helicobacter pylori*, *Lactobacillus plajomi*, exopolysaccharide, antibacterial activity, antioxidant activity

## Abstract

*Helicobacter pylori* is a cause of gastric cancer. We extracted the exopolysaccharide (EPS) of *Lactobacillus plajomi* PW-7 for antibacterial activity versus *H. pylori*, elucidating its biological activity and structural characteristics. The minimum inhibitory concentration (MIC) of EPS against *H. pylori* was 50 mg/mL. Disruption of the cell membranes of pathogenic bacteria by EPS was indicated via the antibacterial mechanism test and confirmed through electron microscopy. EPS also has antioxidant capacity. The IC_50_ of EPS for 2,2-Diphenyl-1-picrylhydrazyl (DPPH) radical, superoxide anions, and hydroxyl radicals were 300 μg/mL, 180 μg/mL, and 10 mg/mL, respectively. The reducing power of EPS was 2 mg/mL, equivalent to 20 μg/mL of ascorbic acid. EPS is a heteropolysaccharide comprising six monosaccharides, with an approximate molecular weight of 2.33 × 10^4^ Da. Xylose had a significant effect on *H. pylori*. EPS from *L. plajomi* PW-7 showed potential as an antibacterial compound and antioxidant, laying a foundation for the development of EPS-based foods.

## 1. Introduction

Fifty percent of the global population have gastric *Helicobacter pylori*. The International Cancer Society identify it as a category I carcinogen. The prevalence of gastric cancer in Asia, particularly East Asia, is significantly higher than the average worldwide [1,2]. *H. pylori* can only survive in gastric acid for a short time. It parasitizes the gastric antrum, resulting in chronic inflammation. The resulting destruction of the mucosal barrier is long lasting and leads to gastritis, gastric and duodenal ulcers, and gastric cancer [3,4]. Current *H. pylori* treatment is mainly based on quadruple antibiotic therapy. However, there are problems with this approach [5]. Antibiotics such as clarithromycin and amoxicillin impact the intestinal flora and have many shortcomings, including varying side effects and antibiotic resistance [6,7]. Caring for a wide range of infected people is problematic. There is thus an unmet medical need for molecules with low antibiotic resistance and reduced side effects able to reduce or prevent *H. pylori* in high-risk groups on a large scale.

Lactic acid bacteria (LAB) are beneficial intestinal microorganisms [8,9]. They are present in many foods, including yogurt, sauerkraut, soy milk, kimchi, and fermented vegetables [10]. LAB have been used in the pharmacy industry to regulate intestinal flora, and even to treat dyspepsia, enteritis, and diarrhea [11]. Moreover, LAB could be used as tanning agents and biodegradable materials in the chemical industry [12]. The literature showed that LAB can assist in the treatment of *H. pylori*. When bismuth quadruple therapy, standard triple therapy, and high-dose PPI Bismuth-containing quadruple therapy were used with LAB, the side effects of antibiotics (diarrhea, constipation, etc.) were reduced and the treatment could be sped up [5,13,14]. In the literature, some lactic acid bacteria have an antibacterial effect on *H. pylori* directly; some studies reported that the supernatant contained the substance that inhibited *H. pylori* [15,16].

Metabolites of LAB, such as EPS, organic acids, bacteriocins, and fatty acids, display beneficial biological activities, such as antibacterial, antioxidant, anticancer, and cholesterol-lowering activities [17,18]. Many studies indicated LAB can inhibit the growth of pathogenic bacteria by secreting acid and bacteriocins [18,19]. LAB exopolysaccharide (EPS) is also receiving increased attention due to the potential health benefits of EPS [20,21,22]. Ryan et al. found that the acid and bacteriocin secreted by *L. salivarius* UCC119 could not inhibit *H. pylori*, suggesting the involvement of other mediators [23]. Liu et al. found that EPS enhanced rUreB-specific protective immunity against *H. pylori* [24]. *L. fermentum* UCO-979C EPS also inhibits *H. pylori* colonization [25]. 

As well as antibacterial activity, LAB EPS also possesses other biological activities, including antioxidant, anti-tumor, and anti-diabetic properties [20,26,27]. It has also found wide application in health care and the food industry. A systematic evaluation of the available literature indicates that EPS can inhibit pathogenic bacteria, although its antibacterial mechanism against *H. pylori* is underexplored. 

We isolated a strain of lactic acid bacteria from Sichuan pickles and found that its supernatant had a strong inhibitory effect on *H. pylori*. Our study aimed to elucidate the *H. pylori*-inhibiting components that are fermented by *L. plajomi* PW-7 and to clarify its activity on *H. pylori.* Furthermore, the mechanism of the antibacterial and antioxidant activity of the substances were investigated.

## 2. Materials and Methods

### 2.1. Materials

*H. pylori* was obtained from the American Type Culture Collection (ATCC). *H. pylori* growth medium consisted of ATCC^®^18 media (trypsin bean soup) and ATCC^®^260 media (tryptic soy agar, 5% defibrated sheep blood (Nanjing maojie microbial technology Co., Ltd., Nanjing, China)) (Qingdao hope bio-technology Co., Ltd., Qingdao, China). A micro aerobic gas bag (Mitsubishi gas chemical Co., Ltd., Tokyo, Japan). *Aeromonas hydrophila*, *Aeromonas salmonicida*, *Bacillus subtilis*, *Escherichia coli*, *Pseudomonas aeruginosa*, *Staphylococcus aureus*, and *Vibrio parahaemolyticus* were provided by the Provincial Marine Resources Development Research Institute (Lianyungang, China) and were cultivated using Luria–Bertani (LB) (Aobox, Beijing, China). *Lactobacillus plajomi* PW-7 was selected from indigenous Sichuan pickles (vegetables: radish, cauliflower, carrot, cowpea, celery, ginger, and green and red pepper with 5% of NaCl) and cultivated in MRS broth (Beijing luqiao technology Co., Ltd., Beijing, China) at 37 °C.

### 2.2. Screening of LAB 

Ten milliliters of Sichuan pickle water was added to MRS broth to form a 3% inoculation, and subsequently cultured at 37 °C for 24 h. The enrichment broth was then diluted, using gradient dilution, using concentrations of 10^−3^, 10^−4^, and 10^−5^. Plates were spread with MRS medium in 50 μL aliquots. Plates were put into an anaerobic tank and incubated for 24 h at 37 °C. The colonies with a calcified zone were selected for subsequent experiments [28].

Selected colonies were inoculated into MRS broth and incubated at 37 °C for 24 h. The fermentation broth was then centrifuged at 8000 rpm for 15 min, and the supernatant was used for Oxford cup bacteriostasis tests, with *H. pylori* as the indicator bacterium; is the test was based on the appearance of transparent circle. Strains inhibiting *H. pylori* were selected and pure strains isolated using streak plates [29].

### 2.3. Morphology of Strains

Colony characteristics: the purified single colony was cultured at 37 °C and observed. The cell morphology was recorded using scanning electron microscopy (SEM, model JFC-1600; JSM-6390LA; JEOL, Tokyo, Japan).

### 2.4. 16S rDNA Gene Identification

The genomic DNA of strains was extracted using a PCR kit. PCR amplification was performed using universal primers 27F (5′-AGAGTTTGATCCTGGCTCAG-3′) and 1492R (5′-GGTTACCTTGTTACGCTT-3′). PCR products were purified and sequenced by Sangon Biotech (Shanghai). NCBI Blast (https://blast.ncbi.nlm.nih.gov, 3 September 2020) was used to analyze the resulting sequences. Phylogenetic trees were drawn using MEGA 6 with Neighbor-Joining [30].

### 2.5. Analysis of Components of Strain PW-7 Inhibiting H. pylori

#### 2.5.1. Hydrogen Peroxide Detection and Elimination

The pH of the fermentation supernatant of strain PW-7 was adjusted to 7.0. Catalase was added, and the final concentration adjusted to 1 mg/mL, before being incubated for 2 h at 37 °C. The solution was then placed in boiling water for 5 min, before the pH was adjusted to 5.0 for hydrogen peroxide detection. Supernatant without catalase treatment (Aladdin, Shanghai, China) was used as a control. *H. pylori* was the indicator bacterium, and 10^7^ CFU/mL suspension was applied to TCC^®^260 medium, cultured at 37 °C for 12 h. An Oxford cup antibacterial test was conducted [31].

#### 2.5.2. Detection and Elimination of Organic Acids

The pH of the PW-7 fermentation supernatant was adjusted to 7.0 using NaOH. Untreated supernatant was used as a control. *H. pylori* was used as the indicator bacteria, and the method was the same as that of 2.5.1 [32].

#### 2.5.3. Bacteriostatic Test of Bacteriocin

The fermentation supernatant was precipitated using ammonium sulfate (20–70%) (Aladdin, Shanghai, China), dialyzed, and then lyophilized (Labconco FreeZon^®^2.5L USA). After Gram-negative and Gram-positive bacteria reached the logarithmic growth phase, 10^7^ CFU/mL bacterial suspension was coated on LB, put on Oxford cup, and PW-7 bacteriocin (100 mg/mL) was added and cultured at 37 °C for 12 h. The culture of *H. pylori* was the same as 2.5.1. The bacteriostatic ability of PW-7 bacteriocin was determined for indicator bacteria using the Oxford cup method [33].

#### 2.5.4. Bacteriostatic Test of EPS

The fermentation supernatant was precipitated for 12 h using 3× ethanol at 4 °C. The precipitate was dissolved, and 4% Trichloroacetic acid (TCA) (Aladdin, Shanghai, China) was added. The resulting solution was left to stand for 30 min before being centrifuged (Thermo Heraeus Multifuge X3R Centrifuge, Eppendorf, Hamburg, Germany), and the supernatant was isolated. As with the bacteriocin antibacterial test, the inhibitory effect of EPS on each indicator bacterium was determined [34].

### 2.6. Structure Characterization of EPS

#### 2.6.1. Monosaccharide Composition Analysis

A chromatographic DionexCarbopacTMPA20 (3 × 150) column (Dionex™ ICS-5000+ TC Thermal Compartment, ThermoFisher, Waltham, MA, USA) was used to determine monosaccharide composition. The mobile phase was: A: H_2_O, B: 15 mM NaOH, C: 15 mM NaOH and 100 mM NaOAc; flow rate: 0.3 mL/min; injection volume: 5 μL; column temperature: 30 °C. Sixteen monosaccharide standards (fucose, rhamnose, arabinose, galactose, glucose, xylose, mannose, fructose, ribose, galacturonic acid, glucuronic acid, galactosamine hydrochloride, glucosamine hydrochloride, N-acetyl-D-glucosamine, gulosaldehyde acid, and mannuronic acid) (Borui sugar Biotechnology Co., Ltd., Yangzhou, China) were dissolved in standard solution. Ten micrograms of polysaccharide were placed in ampoules. An amount of 10 mL of Trifluoroacetic acid (3 M) was added, and the solution then underwent hydrolysis for 3 h at 120 °C. The resulting solution was accurately absorbed, transferred to a tube, and dried under nitrogen. An amount of 10 mL of water was added; before vortexing and being absorbed at 100 μL, 900 μL of deionized water was added, and the solution was centrifuged at 12,000 rpm for 5 min. The supernatant was isolated for ion chromatography (IC) analysis [35].

#### 2.6.2. Determination of Molecular Weight

Polysaccharide molecular weight and purity were determined at 25 °C using high-performance gel permeation chromatography (HPGPC) (Shimadzu LC-10A HPLC, Shimadzu, Kyoto, Japan), with ultrapure water as the mobile phase and the flow rate set to 1.0 mL/min. A 2 mg/mL polysaccharide sample solution was prepared using a disposable microporous filter membrane (0.22 μm). After filtration, 100 μL was injected into the chromatograph. By using the elution peak times and multiplicity, the sample purity and average molecular weight were calculated using GPC software (Agilent, Palo Alto, CA, USA ) [36].

### 2.7. Minimum Inhibitory Concentration (MIC)

Suspensions of *E. coli*, *S. aureus*, and *H. pylori* were diluted gradually. A suspension at 10^7^ CFU/mL was used as a standard. In the experimental group, 2 mL of LB culture solution and then 40 μL of spare bacterial solution were added to test tubes. EPS was added to each tube to reach a series of final concentrations: 5, 10, 20, 30, 40, 50, 60, 70, 80, 90, 100 mg/mL. The control group was prepared similarly, without the addition of EPS. Test tubes were incubated at 37 °C for 24 h. After culture, 100 μL of the bacterial broth was spread on LB plates. Observations were made after 24 h of inverted culture at 37 °C. The EPS concentration of the first plate with a single observable colony was taken as the lowest inhibitory concentration [37].

### 2.8. Antibacterial Mechanism

#### 2.8.1. Cell Membrane Permeability

Using the method of Qiao et al. [38], *E. coli*, *H. pylori*, and *S. aureus* in their logarithmic growth phase were inoculated into LB with an EPS concentration of 1× MIC and 2× MIC at 2% inoculum volume. After mixing, samples were cultured at 37 °C. LB without added samples acted as controls. Samples were measured every 2 h for 8 h. Supernatant was centrifuged at 3500 rpm for 10 min. Absorbance was measured at 450 nm using a full wavelength microplate reader (Multiskan GO, Thermo Scientific, Waltham, MA, USA). A bacterial culture medium without samples was used as the control.

#### 2.8.2. Cell Membrane Integrity

Cell membrane integrity was determined by measuring the release of cell components. *E. coli*, *H. pylori* and *S. aureus* were cultured to the logarithmic phase, then centrifuged at 5000 rpm for 5 min. The precipitate was washed with 0.1 M PBS (pH 7.4) (Aladdin, Shanghai, China) three times before resuspension. To measure protein and reduce sugar concentrations in the supernatant, two milliliters of cell suspension were cultured with blank, 1× MIC, and 2× MIC for 4 h before being centrifuged at 10,000 rpm for 5 min. The released nucleic acid concentration was determined at 260 nm [39].

#### 2.8.3. NPN Uptake

The uptake of N-phenyl-1-naphthylamine (NPN) was determined using the method of Zhang et al. [40]. Briefly, *E. coli*, *H. pylori*, and *S. aureus* in the logarithmic phase were centrifuged at 5000 rpm for 10 min, then washed with 0.5% NaCl three times before being re-suspended to an OD_600nm_ of 0.5. The suspension was treated with different EPS concentrations (1× MIC, 2× MIC) for 1 h. After centrifugation, the suspension was washed and re-suspended using 0.5% NaCl. An amount of 200 mL of bacterial suspension was mixed with 1.5 μL of NPN (100 mM) (Aladdin, Shanghai, China). Fluorescence was measured at 350 nm and 420 nm using a full wavelength multifunctional microplate reader (Infinite M1000 Pro, TECAN, Homrechtikon, Switzerland).

#### 2.8.4. Scanning Electron Microscopy

For the experimental group, 2 mL of LB was added to each test tube, then logarithmic phase broth (2% inoculation), before EPS was added to reach a final EPS concentration of 2× MIC. Samples were cultured at 37 °C for 1 h, then centrifuged at 5000 rpm for 10 min to remove supernatant. For the control group, logarithmic phase broth was centrifuged directly. The precipitate was removed before being washed three times with 0.2 M PBS (pH 7.4) and then fixed with 2.5% glutaraldehyde (Aladdin, Shanghai, China). Mixing occurred overnight. Slides were soaked in 75% ethanol and then allowed to dry. Different gradients of bacterial solution were applied to the slide before the addition of 0.2 M PBS (pH 7.4) followed by spreading, drying, and washing with 30%, 50%, 70%, 80%, 90%, and 100% ethanol as required by SEM [41].

### 2.9. Antioxidant Activity of Exopolysaccharides In Vitro

#### 2.9.1. DPPH Free Radical Scavenging Experiment

The rate of DPPH free radical scavenging was determined using the method of Dong [42]. Two hundred microliters of EPS at different concentrations were added to 200 μL of DPPH (0.1 mM of absolute ethanol) (Aladdin, Shanghai, China). After mixing, absorbance was measured at 517 nm after equilibrating at room temperature for 30 min. Ascorbic acid (Vc) (Aladdin, Shanghai, China) was used as a positive control. DPPH free radical scavenging activity was expressed as:Scavenging ability (%) = [1 − (A_a_ − A_b_)/A_c_] × 100(1)
where A_a_ is the absorbance of the DPPH free radical solution and polysaccharide sample; A_b_ is the absorbance of anhydrous ethanol and polysaccharide; and A_c_ is the absorbance of the DPPH radical solution and absolute ethanol.

#### 2.9.2. Hydroxyl Radical Scavenging Experiment

Amounts of 100 μL of FeSO_4_·7H_2_O (9 mM), 100 μL of salicylic acid anhydrous ethanol solution (9 mM), 100 μL of 0.03% H_2_O_2_ (Aladdin, Shanghai, China), and 100 μL of polysaccharide samples at different concentrations were mixed for 15 m at 37 °C. Absorbance was measured at 510 nm, with water as a blank [43]. Hydroxyl radical scavenging activity was calculated using:Scavenging ability (%) = [(A_c_ − (A_a_ − A_b_))/A_c_] × 100(2)
where A_a_ is the absorbance value of the polysaccharide sample; A_b_ is the absorbance value without H_2_O_2_; and A_c_ was the blank control.

#### 2.9.3. Superoxide Anion Radical Scavenging Experiment

Superoxide anion was modified using the method of Zhang [44]. Two hundred microliters of polysaccharide samples at different concentrations were mixed with 1 mL of Tris HCl (50 mM pH 8.2) and then incubated for 10 min at 25 °C. Thirty microliters of pyrogallol (6 mM) (Aladdin, Shanghai, China) was added. The reaction was then kept in the dark for 30 min. Thirty microliters of HCl (6 mM) was added to terminate the reaction. Absorbance was measured at 320 nm, with water used as a control. The scavenging rate of superoxide anion radical was calculated as:Scavenging activity (%) = [1 − (A_a_ − A_b_)/A_c_] × 100(3)
where A_a_ is the absorbance of polysaccharide; A_b_ is the absorbance without pyrogallol; and A_c_ is the absorbance of the control.

#### 2.9.4. Reducing Power

Amounts of 200 μL of polysaccharide samples with different concentrations, 200 μL of PBS (0.2 M, pH 6.6), and 200 μL of potassium hexacyanoferrate (1%, *w*/*v*) (Aladdin, Shanghai, China) were mixed, then cultured at 50 °C for 20 min, with 200 μL of TCA (10%) added to terminate the reaction. After centrifugation at 5000 rpm for 10 min, 500 μL of supernatant was mixed for 10 min with 100 μL of FeCl_3_ (0.2%) (Aladdin, Shanghai, China) and 2.5 mL of water. Absorbance was measured at 700 nm [44].

### 2.10. Statistical Significance

Experiments were repeated three times. Data are presented as means ± SD. SPSS statistics, version 17.0 for windows, was used for statistical analysis. *p* < 0.05 was the threshold for statistical significance. 

## 3. Results and Discussions

### 3.1. Screening and Identification of Strain

#### 3.1.1. Screening of Strain

Ten strains of LAB were isolated from pickle samples. Four strains displaying an inhibition zone against *H. pylori* were screened further. An optimal strain exhibiting enhanced inhibition was selected. We named this strain PW-7 (Figure 1a).

#### 3.1.2. Morphological Characteristics of Strain PW-7

PW-7 formed small, round, milk-white colonies with moist, smooth edges on MRS medium (Figure 1b). Strain PW-7 was Gram-positive and rod-shaped (Figure 1c).

#### 3.1.3. Sequence Analysis of 16S rDNA of Strain PW-7

The genome of strain PW-7 was extracted, and 16S rDNA was amplified using universal primers and detected using agarose gel electrophoresis (see Figure 1d). The fragment size was approximately 1.5 kb. The resulting PW-7 gene sequence was submitted to GenBank. By using sequence similarity comparison, it was determined that PW-7 belonged to the genus *Lactobacillus*. Sequences from closely related strains were compared using MEGA 7, with the neighbor-joining method used to construct a phylogenetic tree. This showed that PW-7 and *Lactobacillus plajomi* were closely related (Figure 1e). On the basis of our analysis of its cellular morphology, its gene sequence, and its biochemical and physiological characteristics, strain PW-7 was classified as *Lactobacillus plajomi* PW-7.

### 3.2. Analysis of Components of Strain PW-7 Inhibiting H. pylori

When compared to untreated PW-7 fermentation broth, catalase treatment did not decrease the inhibitory effect of the broth, indicating that inhibition was not caused by hydrogen peroxide (see Appendix A). When the pH of the fermentation supernatant was adjusted to 7.0, its antibacterial ability did not decrease, indicating inhibition was not caused by release of acid (Appendix A). Bacteriocins did not inhibit Gram-negative and Gram-positive bacteria (Appendix A), suggesting that bacteriocin was not responsible for the observed antimicrobial activity, and that other bacteriostatic substances may be present. EPS can inhibit Gram-negative and Gram-positive bacteria (Table 1). 

A review of the literature suggested that EPS from LAB can inhibit pathogenic bacteria. EPS from *Lactobacillus plantarum* WLPL04 and EPS from *Lactobacillus rhamnosus* are known to inhibit pathogenic micro-organisms [43,45]. We conclude that the EPS of *L. plajomi* PW-7 is a substance with antibacterial effect.

### 3.3. Structure Analysis of PW-7 EPS

#### 3.3.1. Monosaccharide Composition of EPS

The observed elution peak was 2.0 min for sodium hydroxide and 40 min for sodium acetate. This was compared to standard monosaccharide chromatograms (Figure 2a). PW-7 EPS mainly comprised galactosamine hydrochloride, glucosamine hydrochloride, galactose, glucose, xylose, and glucuronic acid. The molar ratio was 2.804:4.217:1.978:6.304:5.435:1 (Figure 2b, Table 2). Therefore, PW-7 EPS was a heteropolysaccharide. The monosaccharide composition of LAB EPS may affect the biological activity [46,47]. The EPS of *L. plantarum* KX041 consisted of rhamnose, fucose, arabinose, xylose, mannose, glucose, galactose, and galacturonic acid [48]. In contrast, the EPS of *L. helveticus* LZ-R-5 was composed of galactose and glucose [49]. Most LAB EPS comprised glucose, galactose, and mannose among others, and a small amount of EPS contained xylose and glucuronic acid [36].

#### 3.3.2. Molecular Weight of EPS

Three distinct molecular weights of EPS were determined using GPC analysis (see Figure 2c). Peak 1 corresponded to 2.33 × 10^4^ Da, with a peak area of 93.07%. The polydispersity index (PDI, Mw/Mn) of EPS was 1.404. Peak 2 corresponded to 8.76 × 10^4^ Da, with a peak area of 5.445% and an EPS PDI of 1.545. Peak 3 corresponded to 1.45 × 10^6^ Da, with a peak area of 1.483% and an EPS PDI of 1.893. According to Wang et al. [50], PDI, as a measure of the molecular weight distribution of EPS, has a significant impact on the characteristics of EPS. Kieran M Lynch found that the molecular weight of LAB EPS is generally between 10^4^–10^6^ Da [51]. Based on peak 1, strain PW-7 could produce heteropolysaccharides, the molecular weight of which was over 10^4^ Da.

### 3.4. Minimum Inhibitory Concentration (MIC)

The minimum inhibitory concentration of extracellular polysaccharide against *E. coli* was 40 mg/mL. The minimum inhibitory concentration against *S. aureus* and *H. pylori* was 50 mg/mL (see Table 3).

### 3.5. Antibacterial Mechanism

#### 3.5.1. Cell Membrane Permeability

EPS affected the membrane permeability of *E. coli*, *S. aureus,* and *H. pylori* (Figure 3). In the control groups of *E. coli*, *S. aureus*, and *H. pylori*, conductivity measurements increased only slightly with time, perhaps resulting from normal bacterial lysis and cell death [52]. When the EPS concentration was greater than or equal to 1× MIC, conductivity measurements increased immediately, rising rapidly with time and polysaccharide concentration. Growth tended to be flat at 6–8 h, suggesting that EPS destroys the normal structure of bacterial cell membranes. Increased cell permeability indicates the leakage of intracellular components, particularly electrolytes such as K^+^, Ca^2+^, and Na^+^ [40].

#### 3.5.2. Integrity of Cell Membranes

As EPS concentration rose, the release of cell components increased significantly (Table 4). When the concentration of EPS was greater than or equal to 1× MIC, the concentration of cell components (OD_260nm_), reducing sugar, and soluble protein increased with increasing EPS concentration. The leakage of nucleic acids, proteins, and other cell components resulted from a loss of membrane integrity [52]. Pradeepa et al. found that LAB EPS alters negatively-charged proteins and glycoproteins on the bacterial cell surface [53]. EPS may damage *Staphylococcus aureus* cells, leading to protein leakage and cell death.

#### 3.5.3. Cell Membrane NPN Uptake

Fluorescence intensity measurements of *E. coli*, *S. aureus,* and *H. pylori* suspensions increased sharply after 1× MIC and 2× MIC treatment (Figure 3). Cell damage was dose-dependent, consistent with membrane integrity assays. Exposure of bacteria to EPS solution may induce structural changes in the bacterial cell membrane and concomitant functional damage. The principal component of cell membranes is phospholipid, and NPN may combine with phospholipid, resulting in significantly increased fluorescence [40].

#### 3.5.4. Electron Microscopic Observations

In the log phase, 2× MIC EPS was reacted for 1h with indicator bacteria. SEM images indicated untreated *E. coli*, *S. aureus*, and *H. pylori* were regular and complete (Figure 4a–c), with normal cell morphology, an intact cell wall, and undisrupted cell membranes. In contrast, bacteria treated with EPS (Figure 4d–f) were pitted, irregular, and sunken, with altered cell morphology. Some cells were broken, with absent or ruptured membranes. Our results show EPS may seriously disrupt the cell membranes of *H. pylori*, resulting in intracellular content leakage [44].

### 3.6. Antioxidant Activity of EPS

DPPH clearance increased with increasing EPS concentration. The IC_50_ for DPPH removal was 300 μg/mL (Figure 5a). DPPH is a relatively stable lipid free radical, able to inhibit the oxidation of free radicals after the addition of antioxidant. The scavenging of superoxide anion via EPS is concentration-dependent. At an EPS concentration of 180 μg/mL, the scavenging rate of superoxide anion reached the IC50. When the concentration of *Lactobacillus helveticus* MB 2-1 EPS was 4 mg/mL, the superoxide radical scavenging rate was 71.82%. When EPS was 6.83 mg/mL, the DPPH scavenging rate was 50% [24]. When the *Lactobacillus rhamnosus* EPS concentration was 4 mg/mL, the DPPH scavenging rates reached (EPS 114) 37.9% and (EPS 111) 63.4%, while the superoxide radical scavenging rates were 5.9% (EPS 115) and 29.4% (EPS 111). However, the DPPH and superoxide radical scavenging rates of PW-7 EPS were higher [43].

The scavenging rate of polysaccharide to hydroxyl radical was positively correlated with concentration (Figure 5c). When EPS concentration was 10 mg/mL, the hydroxyl radical scavenging rate reached IC_50_. The EPS of strain PW-7 had strong reducing power (Figure 5d) [44].

### 3.7. Effect of Monosaccharide on Cell Membrane Permeability 

The effect on the membrane permeability of each EPS monosaccharide component was evaluated. Galactose had a greater effect on *E. coli* and glucuronic acid had a greater effect on *S. aureus.* Xylose had a stronger effect on *H. pylori* (Figure 6). EPS had more than 7.6% xylose. 

Prior work suggested EPS had an inhibitory effect on *H. pylori*; however, the antibacterial effect of EPS components on *H. pylori* was not further analyzed [24,25]. It was found that the EPS (EPSWLD and EPSMLD) produced by wild-type and mutant *L. delbrueckii* contained monosaccharides such as ribose, xylose, arabinose, rhamnose, fructose, glucose, mannose, and galactose, and the EPS inhibited *Bacillus subtilis* and *S. aureus*. EPS also has antioxidant capacity. When the concentration was 10 mg/mL, EPSWLD and EPSMLD have the highest DPPH activity (73.4% and 65.6%) and H_2_O_2_ scavenging activity (88.5% and 78.6%) [54]. The EPS of *L. plantarum* WLPL04 was composed of xylose, glucose, and galactose, and the molar ratio was 3.4:1.8:1. It can inhibit the pathogenic biofilm of *Pseudomonas aeruginosa*, *E. coli*, *Salmonella typhimurium*, and *S. aureus* [45]. Therefore, we speculated that the EPS (heteropolysaccharides) with a higher portion of xylose might be a factor to inhibit *H. pylori*, which paves the way for future research on inhibiting *H. pylori*.

## 4. Conclusions

To conclude, EPS is a heteropolysaccharide extracted from *L. plajomi* PW-7 with antibacterial activity against *H. pylori, S. aureus*, and *E. coli*. The MIC of the EPS for *H. pylori*, *S. aureus*, and *E. coli* was 50, 50, and 40 mg/mL, respectively. Antibacterial activity exhibited by EPS increases with increasing concentration and time. EPS exerts its antibacterial effect by disrupting the cell membranes of *H. pylori*, *S. aureus,* and *E. coli*. The molecular weight of EPS was approximately 2.33 × 10^4^ Da (93.07%). The EPS from xylose may exert important effects on anti-*H. pylori*. EPS is also a potent antioxidant. To summarize, the EPS from *L. plajomi* PW-7 has clear antibacterial and anti-oxidation properties. The PW-7 strain should be useful in health care and as a food additive, along with a range of other important applications.

## Figures and Tables

**Figure 1 foods-10-02453-f001:**
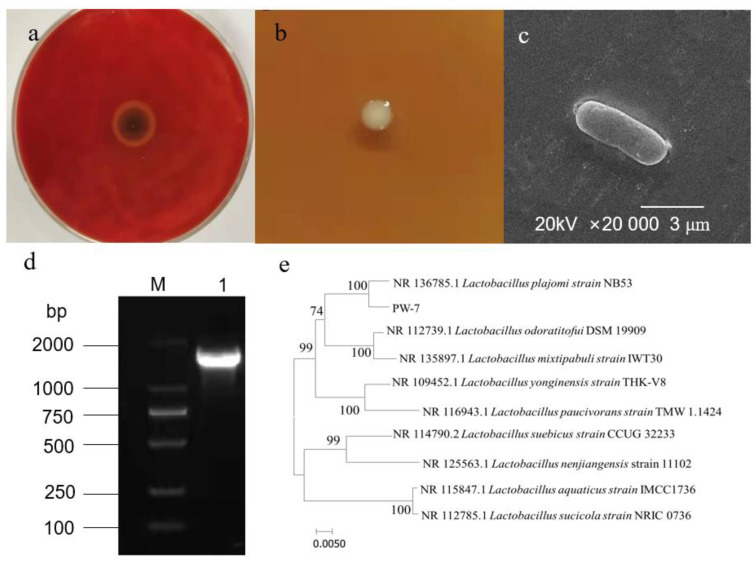
(**a**): Antibacterial effect of PW-7 on *H. pylori*; (**b**): colony morphology of strain PW-7; (**c**): scanning electron microscopy of PW-7 morphology; (**d**): agarose gel electrophoresis of 16S rDNA PCR products; M: DNA marker; 1: strain PW-7 16S rDNA PCR amplification product; (**e**): phylogenetic tree based on 16S rDNA gene sequences.

**Figure 2 foods-10-02453-f002:**
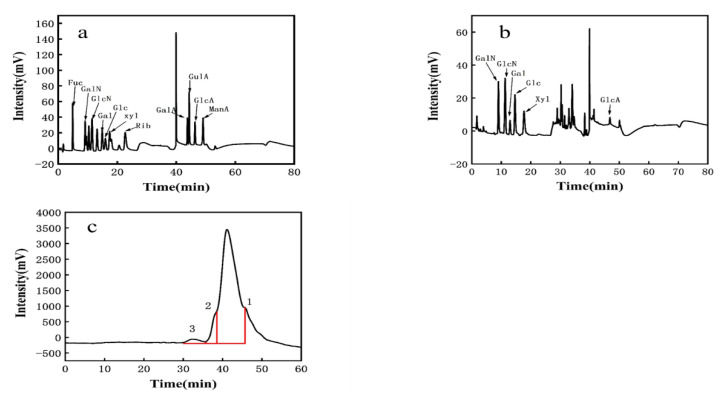
(**a**): The HPLC chromatograms of monosaccharides standard; (**b**): the HPLC chromatograms of PW-7 EPS; (**c**): GPC analysis of EPS.

**Figure 3 foods-10-02453-f003:**
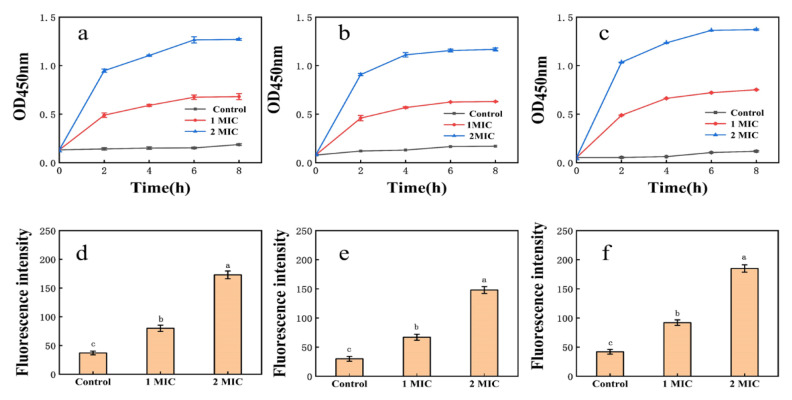
(**a**): Effect of EPS on the membrane permeability of *E. coli*; (**b**): effect of EPS on the membrane permeability of *S. aureus*; (**c**): effect of EPS on the membrane permeability of *H. pylori*; (**d**): NPN uptake of *E. coli* treated with EPS; (**e**): NPN uptake of *S. aureus* treated with EPS; (**f**): NPN uptake of *H. pylori* treated with EPS. Values are expressed as the mean ± SD of triplicates. Different lowercase letters indicate statistically significant differences between means (*p* < 0.05) for EPS of different concentration.

**Figure 4 foods-10-02453-f004:**
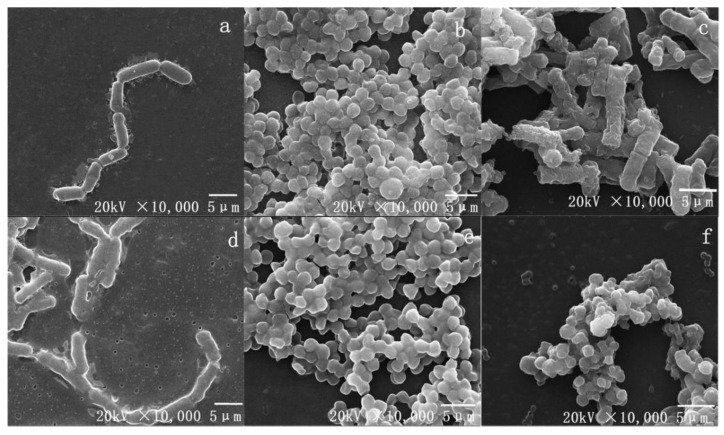
The SEM photography, control: (**a**): *E. coli*, (**b**): *S. aureus*, (**c**): *H. pylori*. Treated with EPS at 2× MIC: (**d**): *E. coli*, (**e**): *S. aureus,* (**f**): *H. pylori*.

**Figure 5 foods-10-02453-f005:**
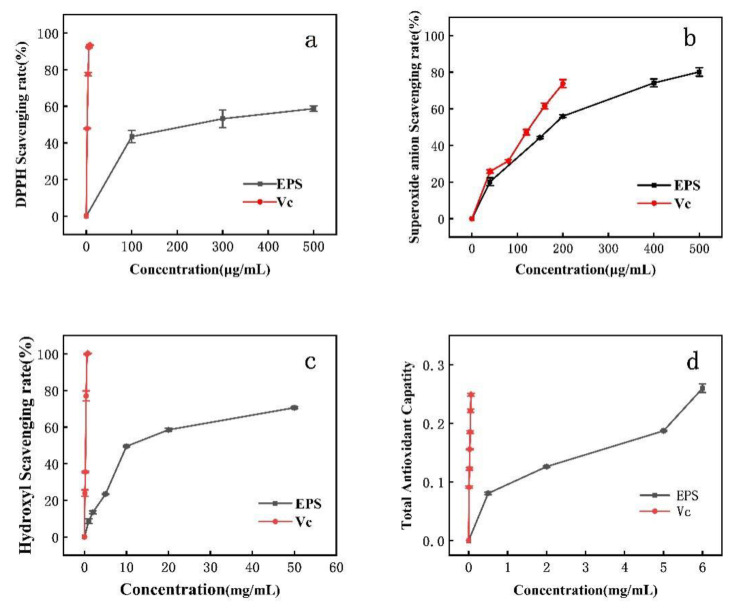
Antioxidant activity of EPS. (**a**): DPPH radical scavenging activity; (**b**): superoxide radical scavenging activity; (**c**): hydroxyl radical scavenging activity; (**d**): reducing power.

**Figure 6 foods-10-02453-f006:**
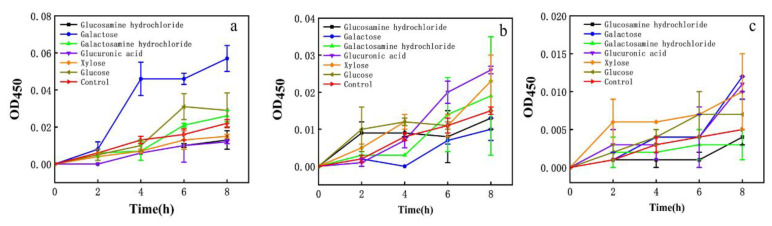
Effect of different monosaccharides on cell membrane impermeability. (**a**) *E. coli*; (**b**) *S. aureus*; (**c**) *H. pylori*.

**Table 1 foods-10-02453-t001:** Antibacterial spectrum of EPS of strain PW-7.

Indicator Bacteria	Diameter of Inhibition Zone (mm)
*A. salmonicida*	31.953 ± 0.497
*H. pylori*	30.613 ± 0.588
*V. parahaemolyticus*	28.727 ± 0.454
*P. aeruginosa*	26.107 ± 0.555
*E. coli*	25.623 ± 0.015
*A. hydrophila*	24.817 ± 0.617
*S. aureus*	24.637 ± 0.791
*B. subtilis*	21.820 ± 0.562

**Table 2 foods-10-02453-t002:** Analysis of monosaccharide composition of PW-7 EPS.

Name	RT (min)	Mole Ratio	Area
Galactosamine hydrochloride	9.025	0.129	10.077
Glucosamine hydrochloride	11.300	0.194	13.538
Galactose	12.959	0.091	3.265
Glucose	14.642	0.290	10.148
Xylose	17.709	0.250	7.630
Glucuronic acid	46.825	0.046	1.404

**Table 3 foods-10-02453-t003:** MIC of EPS against indicator bacteria.

Indicator Bacteria	MIC (mg/mL)
*E. coli*	40
*S. aureus*	50
*H. pylori*	50

**Table 4 foods-10-02453-t004:** Effects of EPS on cell components’ release of *H. pylori*.

Indicator Bacteria	Experience Group	Cell Constituents’ Release		
		Cell Constituents (OD_260nm_)	Soluble Protein (mg/mL)	Reducing Sugar (mg/mL)
*H. pylori*	Control	0.024 ± 0.007 ^d^	0.185 ± 0.005 ^d^	0.043 ± 0.001 ^e^
	1× MIC	0.057 ± 0.006 ^b^	2.043 ± 0.057 ^c^	0.400 ± 0.016 ^c^
	2× MIC	0.115 ± 0.008 ^a^	3.312 ± 0.059 ^a^	0.858 ± 0.009 ^d^
*E. coli*	Control	0.037 ± 0.008 ^cd^	0.252 ± 0.023 ^d^	0.039 ± 0.002 ^e^
	1× MIC	0.066 ± 0.009 ^b^	2.260 ± 0.017 ^b^	1.148 ± 0.050 ^b^
	2× MIC	0.123 ± 0.007 ^a^	3.385 ± 0.054 ^a^	1.408 ± 0.057 ^a^
*S. aureus*	Control	0.019 ± 0.001 ^d^	0.169 ± 0.001 ^d^	0.039 ± 0.001 ^e^
	1× MIC	0.052 ± 0.004 ^bc^	1.960 ± 0.048 ^c^	1.098 ± 0.149 ^b^
	2× MIC	0.107 ± 0.002 ^a^	3.295 ± 0.017 ^a^	1.429 ± 0.011 ^a^

Values represent means of three independent replicates ± SD. Different lowercase letters within the same column indicate statistically significant differences between treatments (*p* < 0.05).

## Data Availability

The interaction data used to support the findings of this study are included within the article and the supporting information file. Also, all the data used to support the findings of this study are available from the corresponding author upon request.

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
