# Peer review of "Anti-Helicobacter pylori Activity of a Lactobacillus sp. PW-7 Exopolysaccharide"

_foods, 2021, doi:10.3390/foods10102453_

Round 1

Reviewer 1 Report

The paper “Anti-Helicobacter pylori Activity of a Lactobacillus sp. PW-7 2 Exopolysaccharide” by Hu et al. describes extraction and characterization of an exopolysaccharide (EPS) from Lactobacillus plajomi and its use as antibacterial against H. pylori.

The idea that  EPS has antibacterial and antioxidant activity is not new. See, for example, few very recent papers (but many others are present in the literature):

Antibacterial potential of a novel Lactobacillus casei strain isolated from Chinese northeast sauerkraut and the antibiofilm activity of its exopolysaccharides.

Xu X , Peng Q , Zhang Y , Tian D , Zhang P , Huang Y , Ma L , Dia VP , Qiao Y , Shi B .

Characterization, the Antioxidant and Antimicrobial Activity of Exopolysaccharide Isolated from Poultry Origin Lactobacilli.

Rajoka MSR, Mehwish HM, Hayat HF, Hussain N, Sarwar S, Aslam H, Nadeem A, Shi J.

Antibacterial and Antibiofilm Activity of Temporin-GHc and Temporin-GHd Against Cariogenic Bacteria, Streptococcus mutans.

Zhong H, Xie Z, Wei H, Zhang S, Song Y, Wang M, Zhang Y.

Characterization and antioxidant activity of an acidic exopolysaccharide from Lactobacillus plantarum JLAU103.

Min WH, Fang XB, Wu T, Fang L, Liu CL, Wang J.

Characterization, the Antioxidant and Antimicrobial Activity of Exopolysaccharide Isolated from Poultry Origin Lactobacilli.

Rajoka MSR, Mehwish HM, Hayat HF, Hussain N, Sarwar S, Aslam H, Nadeem A, Shi J

Characterization and Antioxidant Activity of an Exopolysaccharide Produced by Rigidoporus microporus (Agaricomycetes).

Jia X, Qu L, Panpan R, Liu S, Wu Y, Xu C.

Characterization and Antioxidant Activity of the Exopolysaccharide Produced by Bacillus amyloliquefaciens GSBa-1.

Zhao W, Zhang J, Jiang YY, Zhao X, Hao XN, Li L, Yang ZN.

and many others …

In addition, this specific EPS presents antibacterial activity, but it is not specific for H. pylori: it has a similar effect on more or less all the bacteria tested (see Table 1 and Figure 3). This effect is possibly due to the quite high minimum concentration, around 40-50 mg/mL.

The results are significant in an in vitro test in a laboratory, but the idea that this compound can be an effective antibacterial against H. pylori  in food is unrealistic. I cannot see how this concentration of EPS can be realized in vivo, where H. pylori aderes to the epitelial cell of the stomach protected by the mucus layer.

Reviewer 2 Report

The aim of the manuscript  was to estimate the activity  of a exopolysaccharide produced by Lactobacillus sp. PW-7 against Helicobacter pylori. The research topic covered in the manuscript is interesting and of public health importance. The layout of the article is correct, below I have included detailed comments on individual sections:
*the use of abbreviations in the abstract may be incomprehensible to readers 
*Line-43: 'The treatment group with LAB could reduce the side effects of antibiotics' - what do authors mean by treatment grup?
*Line 45: 'Both LAB and the supernatant of LAB could inhibit H. pylori'  - I would like authors to explain if all LAB species have the ability to inhibit  th H. pylori growth
*Line47: '... are beneficial' - bedenficial in which terms?
*Line 59-64: Has the research the authors are writing about been published? If not, I suggest moving the part about preliminary research to the methodology, and here clearly defining the purpose of the research.
Line 61-62: 'we found that the antibacterial substance was EPS' - how did the authors arrive at this conclusion?
*general comment on the methodology - please be more precise in describing the methodology, specify the manufacturers of the materials used

Round 2

Reviewer 1 Report

See my comment below